# Longitudinal Asthma Patterns in Italian Adult General Population Samples: Host and Environmental Risk Factors

**DOI:** 10.3390/jcm9113632

**Published:** 2020-11-11

**Authors:** Sara Maio, Sandra Baldacci, Marzia Simoni, Anna Angino, Stefania La Grutta, Vito Muggeo, Salvatore Fasola, Giovanni Viegi

**Affiliations:** 1Pulmonary Environmental Epidemiology Unit, CNR Institute of Clinical Physiology (IFC), 56126 Pisa, Italy; baldas@ifc.cnr.it (S.B.); marzia_simoni@libero.it (M.S.); anginoa@ifc.cnr.it (A.A.); giovanni.viegi@irib.cnr.it (G.V.); 2CNR Institute for Biomedical Research and Innovation (IRIB), 90146 Palermo, Italy; stefania.lagrutta@irib.cnr.it (S.L.G.); salvatore.fasola@irib.cnr.it (S.F.); 3Department of Economics, Business and Statistics, University of Palermo, 90128 Palermo, Italy; vito.muggeo@unipa.it

**Keywords:** asthma, epidemiology, cohort, latent transition analysis, comorbidities, smoke, vehicular traffic

## Abstract

Background: Asthma patterns are not well established in epidemiological studies. Aim: To assess asthma patterns and risk factors in an adult general population sample. Methods: In total, 452 individuals reporting asthma symptoms/diagnosis in previous surveys participated in the AGAVE survey (2011–2014). Latent transition analysis (LTA) was performed to detect baseline and 12-month follow-up asthma phenotypes and longitudinal patterns. Risk factors associated with longitudinal patterns were assessed through multinomial logistic regression. Results: LTA detected four longitudinal patterns: persistent asthma diagnosis with symptoms, 27.2%; persistent asthma diagnosis without symptoms, 4.6%; persistent asthma symptoms without diagnosis, 44.0%; and ex -asthma, 24.1%. The longitudinal patterns were differently associated with asthma comorbidities. Persistent asthma diagnosis with symptoms showed associations with passive smoke (OR 2.64, 95% CI 1.10–6.33) and traffic exposure (OR 1.86, 95% CI 1.02–3.38), while persistent asthma symptoms (without diagnosis) with passive smoke (OR 3.28, 95% CI 1.41–7.66) and active smoke (OR 6.24, 95% CI 2.68–14.51). Conclusions: LTA identified three cross-sectional phenotypes and their four longitudinal patterns in a real-life setting. The results highlight the necessity of a careful monitoring of exposure to active/passive smoke and vehicular traffic, possible determinants of occurrence of asthma symptoms (with or without diagnosis). Such information could help affected patients and physicians in prevention and management strategies.

## 1. Introduction

Asthma prevalence has reached epidemic proportions (1–18%) [1] due to host and environmental risk factors [2,3].

According to the Global Burden of Disease Study, in 2017, total deaths from asthma were 495,000 globally [4]; all-age prevalent cases of asthma were 273 million and all-age incident cases were 43 million (about 50% and 70% of all the chronic respiratory diseases, respectively) [5].

Asthma may begin at any age (mainly in children). It may clinically persist, conclusively remit, or present combination of remissions and relapses over time [6,7,8]; as a result, its course is difficult to characterize and its prognosis difficult to predict [7]. In adults, asthma that has persisted from childhood is potentially difficult to treat and it is a distinct clinical phenotype; however, characteristics of these patients are not well documented [9]. A thorough characterization of adult patients with persistent asthma and their risk factors may help clinicians in establishing treatment plans and applying preventive interventions [9,10]. Indeed, from a clinical perspective, it is essential to elucidate the asthma natural history and long-term outcomes; studies on asthmatic cohorts can help [6].

Moreover, although asthma control can be achieved in the majority of patients participating in controlled trials, available data show that this is not the case in real-life [11]; thus, it becomes important to deepen the knowledge in this setting.

Disentangling respiratory diseases phenotypes is a current research challenge [8,12,13], and “unsupervised” or “data-driven” approaches have been proposed. These approaches may allow defining objective, novel, or previously unrecognized phenotypes, by using clustering algorithms accounting for multiple disease features [14,15]. In particular, latent transition analysis (LTA) is a statistical method that can incorporate the longitudinal patterns of several disease manifestations into a comprehensive statistical model, which simultaneously defines phenotypes and their changes over time [12,14]. When disease prevalence changes over time, transition probabilities can help explain the dynamics of such change [14]. However, most studies on asthma phenotyping using data-driven methods involve patients with moderate to severe asthma and/or clinical settings, which limits the possibility of generalizing the findings to the general population [15].

In this framework, data of the AGAVE survey (“Severe Asthma: epidemiological and clinical cohorts follow up by registry and questionnaires; therapeutic appropriateness and outcome assessment, according to GINA guidelines”) were analyzed.

The AGAVE survey, funded by the Italian Medicines Agency (AIFA), was carried out during 2011–2014 to assess asthma modifiable risk factors and the effectiveness of therapeutic strategies, in epidemiological and clinical samples, through the implementation of an online registry [16].

This manuscript focuses on the AGAVE epidemiological sample, with the aim of assessing cross-sectional asthma phenotypes, their longitudinal patterns, and the associated risk factors.

## 2. Materials and Methods

### 2.1. Study Population

A randomized general population sample, living in the rural Po Delta area, North Italy, was involved in two subsequent cross-sectional surveys: first survey (1980–1982), 3284 subjects, and second survey (1988–1991), 2841 subjects. In total, 2136 subjects participated in both surveys [17].

The same protocol and selection method were used to enroll a random general population sample living in the urban and suburban area of Pisa, Central Italy, involved in three subsequent cross-sectional surveys: first survey (1985–1988), 3865 subjects; second survey (1991–1993), 2841 subjects; and third survey (2009–2011), 1620 subjects [18]. Overall, 2257 subjects participated in both the first and the second surveys, 1107 subjects in both the second and the third surveys, and 849 in all three Pisa surveys [18].

In the AGAVE survey (2011–2014), all subjects reporting asthma diagnosis or asthma symptoms (asthma attacks or wheezing) in any of the previous epidemiological surveys were invited: 68% agreed to participate (*n* = 668). In total, 454 subjects were investigated at both AGAVE baseline and follow-up. In this manuscript, only subjects having a baseline age of ≥16 years were taken into account (*n* = 452) (Figure 1).

### 2.2. Data Collection Tool

Each subject underwent a telephone interview lasting about 20 min, based on a questionnaire covering the main items of the GINA guidelines [1], which include asthma symptoms, treatment, exacerbation, symptom control, comorbidity, and exposure to risk factors. The questions were derived from different validated questionnaires such as the European Community Respiratory Health Survey (ECRHS) questionnaire [19] and those used by our research group in other surveys about respiratory health [20,21,22]. The latter were derived from the National Heart Lung and Blood Institute (NHLBI, Bethesda, MD, USA) questionnaire. The AGAVE questionnaire was reviewed and approved by an interdisciplinary internal board, comprised of pulmonologists, allergists, and epidemiologists.

The AGAVE epidemiological questionnaire investigates: asthma clinical history, asthma diagnosis, and use of health services due to asthma throughout life; asthma symptoms, comorbidities, exacerbations, and use of health services due to asthma in the last 12-months; asthma symptoms, asthma control, and asthma treatment in the last month; and current exposure to risk factors.

The same questionnaire was used in the first and second interviews allowing to collect the answers to repeated questions in a longitudinal fashion.

An extract of the AGAVE epidemiological questionnaire, reporting the questions used to define the variables in this manuscript, is available in the Appendix A.

The AGAVE study protocol, patient information sheet, and consent form were approved by the Ethics Committee of the Pisa University Hospital (Prot. No. 17658, 21 March 2011).

### 2.3. Statistical Analyses

Statistical analyses were carried out using the Statistical Package for the Social Sciences (SPSS version 26.0) and the R software (version 3.5.1). Comparisons among groups were performed by Chi-square test for categorical variables and analysis of variance for continuous variables. The significance level was set at 0.05.

Post-hoc analyses were run to assess the sources of statistically significant results, using adjusted standardized residuals for contingency tables larger than the 2 × 2.

LTA was performed using the R package CAT_LVM (Version 0.9.0 alpha, available at https://msu.edu/~chunghw/downloads.html). The main advantage of LTA, over other clustering techniques, is its ability to incorporate into a comprehensive statistical model information about possible changes in the disease characteristics over time, in a longitudinal fashion. Indeed, differently from cross-sectional analytical approaches such as latent class analysis [23], LTA requires temporal variability in all the variables and is much more indicated for characterizing transitions over time [24]. LTA detects unobservable (latent) subgroups (“classes” or “phenotypes”) of subjects based on the values of multiple observed (or “manifest”) variables. The latent classes are exhaustive and mutually exclusive, and they are not assumed to be stable over time. Indeed, LTA also estimates the probabilities of transitions from one latent class to another between different time points [25].

The characterization of asthma (cross-sectional) phenotypes was based on the presence/absence of asthma outcomes, measured at AGAVE baseline and 12-month follow-up (manifest variables), defined as follows: previous physician diagnosed asthma, if the subject answered “YES, but I no longer have it” to the question “Has your doctor ever told you that you have bronchial asthma?”; current physician diagnosed asthma, if the subject answered “YES, I still have it” to the question “Has your doctor ever told you that you have bronchial asthma?”; current asthma attacks if the subject answered “YES” to the question “During the past 12 months, have you had attacks of shortness of breath with wheezing or whistling, apart from common colds?”; and current wheeze if the subject answered “YES” to the question “During the past 12 months, have you had wheezing or whistling, apart from common colds?”.

The model with the lowest Bayesian Information Criterion (BIC), i.e., associated with the best balance of model fit and parsimony, was selected; at each time point, the subjects were assigned to the phenotype associated with the maximum posterior probability of latent class membership [25]. Thus, the longitudinal patterns were defined based on the observed phenotype transitions.

Host and environmental risk factors associated with the longitudinal asthma patterns were assessed through multinomial logistic regression. Only significant variables were retained in the regression analysis to obtain a parsimonious model and improve the statistical power.

Sensitivity analyses were performed comparing AGAVE asthma phenotypes to those of the previous epidemiological surveys from which AGAVE subjects were selected.

## 3. Results

### 3.1. Baseline Subject Characteristics

In total, 452 subjects aged ≥16 years, and participating in both the interviews, were included in the analyses (Table 1). As regards the time frame, the target was 12 months between the first and second questionnaires within a study period of 24 months. The actual result was a mean interval of 15 ± 4 months between the two interviews.

The mean age was 56.7 years; most subjects were females (52.7%), with a middle-low educational level. Fifty-one percent of subjects were overweight–obese (Table 1).

About 75% of the subjects reported skin prick test positivity and about 40% family history of asthma (Table 2).

The most frequent asthma comorbidity was allergic rhinitis (40.8%), followed by gastroesophageal reflux disease (GERD) (29.6%), sleep apnea, recurrent respiratory infections, and chronic obstructive pulmonary disease (COPD) (about 13–14%); very few reported nasal polyps (Table 2).

Over 50% of subjects were exposed to vehicular traffic near home, 40.3% were ex-smokers, and 19.4% current smokers. Less than 20% of subjects were exposed to secondhand smoke (Table 2).

### 3.2. Asthma Phenotypes

Three cross-sectional phenotypes were detected by LTA and labeled as: “asthma diagnosis and current asthma symptoms” due to the very high probability of asthma symptoms (from 50.4% to 73.8%) and diagnosis (about 100%) in the model; “current asthma symptoms” due to the high probability of asthma symptoms (from 10% to 32%) and very low probability of asthma diagnosis (0–1.5%); and “previous asthma diagnosis” due to the low probability of symptoms (0–12%) and very high probability of previous asthma diagnosis (about 100%). The most frequent phenotype was “current asthma symptoms” at both baseline and follow-up (about 45%), followed by “asthma diagnosis and current asthma symptoms” at baseline and “previous asthma diagnosis” at follow-up (Table 3).

The transition plot showed a high stability of phenotypes from baseline to follow-up, ranging from 86.8% for “asthma diagnosis and current asthma symptoms” to 96.6% for “current asthma symptoms”. In addition, 93.2% of subjects showed a long-term asthma remission, reporting no more asthma symptoms or diagnosis with respect to the previous epidemiological studies: this indicates an elevated stability of “previous asthma diagnosis” cross-sectional phenotype (Figure 2).

Indeed, 13.2% of subjects showed an improvement of asthma status (from “asthma diagnosis and current asthma symptoms” to “previous asthma diagnosis”) and 6% a worsening of asthma status in the last 12 months (from “previous asthma diagnosis” to “asthma diagnosis and current asthma symptoms”). At last, 1.5% of subjects reported a new asthma diagnosis in the last 12 months (from “current asthma symptoms” to “asthma diagnosis and current asthma symptoms”) (Figure 2).

Based on the transition plot (Figure 2), the following longitudinal patterns were defined: “persistent asthma diagnosis with persistent/incident asthma symptoms”, i.e., subjects with asthma diagnosis reporting asthma symptoms (wheeze or asthma attacks) at both baseline and follow-up or reporting new asthma symptoms at follow-up (27.2%); “persistent asthma diagnosis with remittent asthma symptoms”, i.e., subjects with asthma diagnosis reporting asthma symptoms at baseline but not at follow-up (4.6%); “persistent asthma symptoms without asthma diagnosis”, i.e., subjects reporting only asthma symptoms, without lifetime asthma diagnosis, at both baseline and follow-up (44.0%); and “ex-asthma”, i.e., subjects reporting “previous asthma diagnosis” at both baseline and follow-up (24.1%) (Figure 3).

### 3.3. Subject Characteristics by Asthma Patterns

Table 4 summarizes the percentage distribution of subject characteristics by longitudinal asthma patterns. Only statistically significant results according to the post-hoc analysis are described.

Subjects in the pattern “persistent asthma diagnosis with remittent asthma symptoms” showed a younger age and a high educational level; subjects in the pattern “persistent asthma symptoms without asthma diagnosis” exhibited older age, a low educational level and overweight; and “ex-asthma” subjects had a medium educational level and normal weight (Table 4).

### 3.4. Asthma-Related Indicators by Longitudinal Asthma Patterns

Subjects in the pattern “persistent asthma diagnosis with persistent/incident asthma symptoms” had their first diagnosis and first asthma symptoms at older age with respect to the other asthma patterns (about 26 years of age vs. 9–13 years) and a higher percentage of family history of asthma with respect to those in the pattern “persistent asthma symptoms without asthma diagnosis” (56.1% vs. 32.6%); in addition, subjects in the pattern “persistent asthma diagnosis with persistent/incident asthma symptoms” more frequently reported last 12-month exacerbations (15.6%) (Table 5).

### 3.5. Baseline Host and Environmental Risk Factors for Longitudinal Asthma Patterns

Subjects in the pattern “persistent asthma diagnosis with persistent/incident asthma symptoms” showed a higher percentage of asthma comorbidities with respect to the other asthma patterns, except for allergic rhinitis that was higher in the pattern “persistent asthma diagnosis with remittent asthma symptoms” (Table 6).

Subjects in the pattern “persistent asthma diagnosis with persistent/incident asthma symptoms” showed a higher percentage of traffic exposure at home address and subjects in the pattern “persistent asthma symptoms without asthma diagnosis” a higher percentage of active and passive smoke exposure (Table 6).

Table 7 describes the results of the multinomial logistic regression analysis for the associations among exposures to host and environmental risk factors and longitudinal asthma patterns (with “ex -asthma” as reference category).

Allergic rhinitis was significantly associated with a three-fold higher risk of having persistent asthma diagnosis with respect to ex-asthma. COPD was significantly associated with “persistent asthma diagnosis with persistent/incident asthma symptoms” (OR 4.76). Sleep apnea was significantly related to “persistent asthma diagnosis with persistent/incident asthma symptoms” and “persistent asthma symptoms without asthma diagnosis” with a 5–6-fold higher risk (Table 7).

Concerning the environmental factors, traffic exposure near home was significantly related to “persistent asthma diagnosis with persistent/incident asthma symptoms” (OR 1.86) and active smoke to “persistent asthma symptoms without asthma diagnosis” (OR 6.24). Passive smoke was significantly associated with “persistent asthma diagnosis with persistent/incident asthma symptoms” and “persistent asthma symptoms without asthma diagnosis” with a 2.6–3.3-fold higher risk (Table 7).

Finally, significant associations were found among: increasing age and higher risk of having asthma symptoms (with/without diagnosis); overweight and a 4.6- and 2.4-fold higher risk of having “persistent asthma diagnosis with remittent asthma symptoms” and “persistent asthma symptoms without asthma diagnosis”, respectively; and family history of asthma and higher risk of “persistent asthma diagnosis with persistent/incident asthma symptoms” (OR 2.15) (Table 7).

## 4. Discussion

Using an unsupervised approach (LTA), we detected four main longitudinal asthma patterns in a sample of adult general population: “persistent asthma diagnosis with persistent/incident asthma symptoms” (27.2%), “persistent asthma diagnosis with remittent asthma symptoms” (4.6%), “persistent asthma symptoms without asthma diagnosis” (44.0%), and “ex-asthma” (24.1%). These patterns were related to different host and environmental risk factors.

### 4.1. Comorbidities

Subjects with allergic rhinitis had a three-fold significantly higher risk of having persistent asthma diagnosis with respect to “exasthma”, highlighting the importance of active management of allergic comorbidities in asthma patients because they may contribute to symptoms burden and poor asthma control, as reported by the international guidelines [1]. Asthma and allergic rhinitis share a similar inflammatory process and nasal allergen exposure in patients with allergic rhinitis yields a generalized airway inflammation including lower airways [1,10].

“Persistent asthma diagnosis with persistent/incident asthma symptoms” was significantly related to COPD (OR 4.76). Asthma and COPD are strongly related, even if they are heterogeneous diseases with various underlying mechanisms. There is broad agreement that patients with features of both diseases have frequent exacerbations, poor quality of life and a more rapid decline in lung function and high mortality [1]. In a US general adult population sample (≥65 years) with active asthma, patients with COPD had a four-fold higher risk of having asthma-related hospitalizations in the last 12 months [26]. A similar result was found in a Canadian study, using health administrative databases: higher rates of asthma claims were found in patients with than in those without COPD [27]. Moreover, active asthma was significantly associated with an increased risk for chronic bronchitis, emphysema, and COPD over a 20-year follow-up [28].

Sleep apnea was significantly related to “persistent asthma diagnosis with persistent/incident asthma symptoms” and “persistent asthma symptoms without asthma diagnosis” with a 5–6-fold higher risk (OR 5.99 and OR 5.32, respectively). Asthma and sleep apnea share many predisposing and aggravating factors; asthma is often accompanied by snoring and apnea, and sleep apnea often combines with asthma. The two diseases present many similar features [29,30] and recent studies have shown that asthma and sleep apnea share a bidirectional relationship where each disorder adversely influences the other one [30,31]. As reported in a US study, unrecognized sleep apnea could be a reason for persistent asthma symptoms during the day and the night [32]. On the other side, the presence of a diagnosed disease with similar clinical manifestations, such as sleep apnea, might lead to lack of recognition or misinterpretation of asthma symptoms, justifying the strong relationship found with persistent asthma symptoms without lifetime asthma diagnosis [29].

### 4.2. Environmental Risk Factors

Reported traffic exposure near home was significantly associated with “persistent asthma diagnosis with persistent/incident asthma symptoms” (OR 1.86 vs. “ex-asthma”). Urban living, characterized by high concentration of air pollutants emitted by vehicular traffic, is an important risk factor for asthma onset and exacerbations [33,34,35]. Recent studies showed a higher risk of persistent asthma in ≥45-year-old subjects living <200 m from a major road (OR 5.21) [36] and a higher risk of last 12-month asthma attacks (OR 1.35) or wheezing (OR 1.24) in adult subjects exposed to moderate/heavy traffic near home [37]. Moreover, the current data are in line with our previous observations that urban living, with respect to rural living, is associated with several adverse effects: e.g., larger prevalence of respiratory symptoms/diseases [17] and higher bronchial hyper-responsiveness, which is an important marker of active asthma [38].

A significantly higher risk of “persistent asthma symptoms without asthma diagnosis” was found in current smokers (OR 6.24), which might indicate underdiagnosis of asthma in this group [39,40].

This relationship emerged also in UK and US studies where active smoke was a risk factor for undiagnosed wheeze at 18 years old (OR 2.54 and OR 2.60, respectively) [39,41]. In a recent paper about the same UK sample, subjects identified as “undiagnosed-wheezers” phenotype had the highest prevalence of smoking habit (74.6%) with respect to the other “wheezers” phenotypes [42]. A possible explanation is that respiratory symptoms, such as wheeze or breathlessness, are misinterpreted as due to smoking, rather than referred to asthmatic condition and, thus, not reported to the physician. On the other hand, asthma-like symptoms are also common in subjects with COPD, which is often smoke-related, making it difficult to disentangle the relationship between smoking and asthma-like symptoms. Smoking plays an important role in asthma management, since asthmatic smokers have worse clinical outcomes compared with nonsmokers, i.e., increased morbidity and mortality, higher frequency of exacerbations, reduced lung function, and worse quality of life [43,44].

Finally, in our study, a strong relationship was found between “persistent asthma diagnosis with persistent/incident asthma symptoms” or “persistent asthma symptoms without asthma diagnosis” and secondhand smoke (OR 2.64 and OR 3.28, respectively). Respiratory health effects of involuntary smoking among children/adolescents are well documented, but fewer studies took into account adult subjects. A study performed on an Italian sample of nonsmoker women showed that passive smoke exposure both to husband and at work resulted a significant risk factor for asthma symptoms (OR 1.71 for recent wheeze and OR 1.85 for recent attacks of shortness of breath with wheeze) [45]. Similar results were found in a Danish study on an adult sample showing that persons exposed to passive smoke were at increased risk of wheeze (OR 1.69) and decreased lung function [46].

### 4.3. Limitations and Strengths

A limitation of this study is the use of a questionnaire for collecting data on respiratory symptoms/diseases, potentially affected by a reporting bias, as it relies upon individual memory. Thus, the estimates of “underdiagnosed” asthma (persistent asthma symptoms without lifetime diagnosis) might be biased, particularly in older subjects, because elderly people with an early onset of the disease may more frequently forget that they had a physician diagnosis, differently from younger people with a more reliable recall. As a consequence, this differential recall bias might have affected the estimates of the time trends and the assessment of the determinants of asthma patterns [47]. Nevertheless, the standardized questionnaire is one of the main investigation tools in respiratory epidemiology [48] and our questions were derived from validated international questionnaires, which already had passed the scrutiny of independent reviewers.

A participation rate of 68% was obtained in the AGAVE study. Comparing the characteristics of subjects included in the AGAVE survey vs. those lost to follow-up, few differences were found, with a lower mean age and more females in the AGAVE sample. No difference was found when considering the enrollment criteria (percentage of asthma symptoms and/or diagnosis in the previous surveys). Moreover, the same cross-sectional asthma phenotypes that emerged in the AGAVE survey were found in the previous surveys, although with different proportions (Appendix A). Thus, the AGAVE sample can be considered representative of the previous samples from which it was extracted according to the enrollment criteria.

The strength of our study is to have analyzed risk factors related to adult asthma patterns in a real-life setting using a multidimensional and unsupervised data analysis approach (the latent transition analysis) which has the advantage of being free from a priori assumptions. Indeed, the choice of the variables to be included in the model was warranted by their clinical relevance, representing the main dimensions of asthma, even if the choice is subjective and may condition the obtained classes [49].

LTA provides innovative perspectives to epidemiological studies. Indeed, while the standard approaches focus on true dichotomous outcomes (disease present or absent), LTA focus on the concept of phenotype, i.e., a combination of several clinical presentations that may occur or not with a given probability. Moreover, LTA is able to describe phenotype changes over time [14].

Finally, it is worth mentioning that a follow-up of these patients will be performed in a new project, starting in the next months, in order to assess the clinical meaning of the detected phenotypes and their relevance for disease prognosis, with special focus on the multimorbidity condition.

## 5. Conclusions

This study about adult asthmatic cohort showed that asthma patterns were differently associated with comorbidities and environmental risk factors exposure. It highlights the necessity of a careful monitoring of exposure to active and passive smoke and to vehicular traffic near the house of residence in the general population, which are possible determinants of occurrence of asthma symptoms (with or without lifetime diagnosis). The increased awareness of these relevant factors and the inclusion of such information in current guidelines could be useful to patients and healthcare providers in prevention and management strategies.

## Figures and Tables

**Figure 1 jcm-09-03632-f001:**
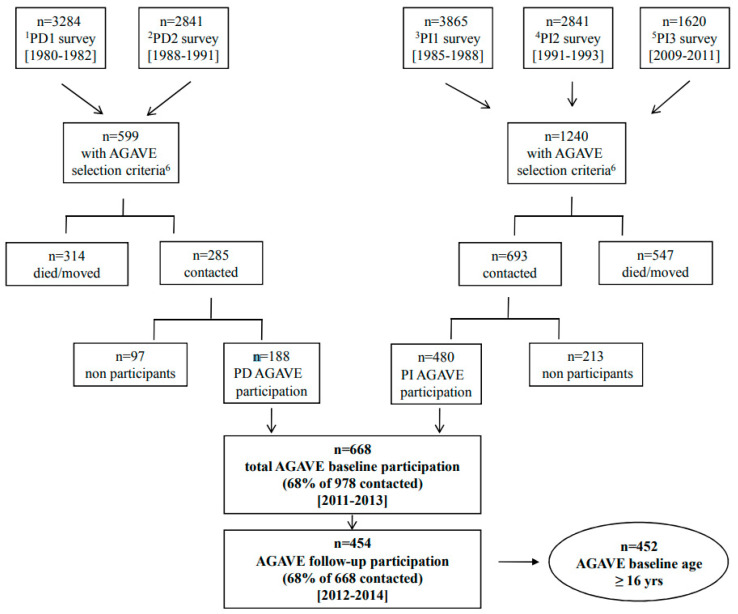
Flow-chart of subjects participation. ^1^PD1, Po Delta first survey; ^2^PD2, Po Delta second survey; ^3^PI1, Pisa first survey; ^4^PI2, Pisa second survey; ^5^PI3, Pisa third survey; ^6^AGAVE selection criteria: subjects reporting asthma diagnosis or asthma symptoms (asthma attacks or wheezing) in at least one of the previous epidemiological surveys.

**Figure 2 jcm-09-03632-f002:**
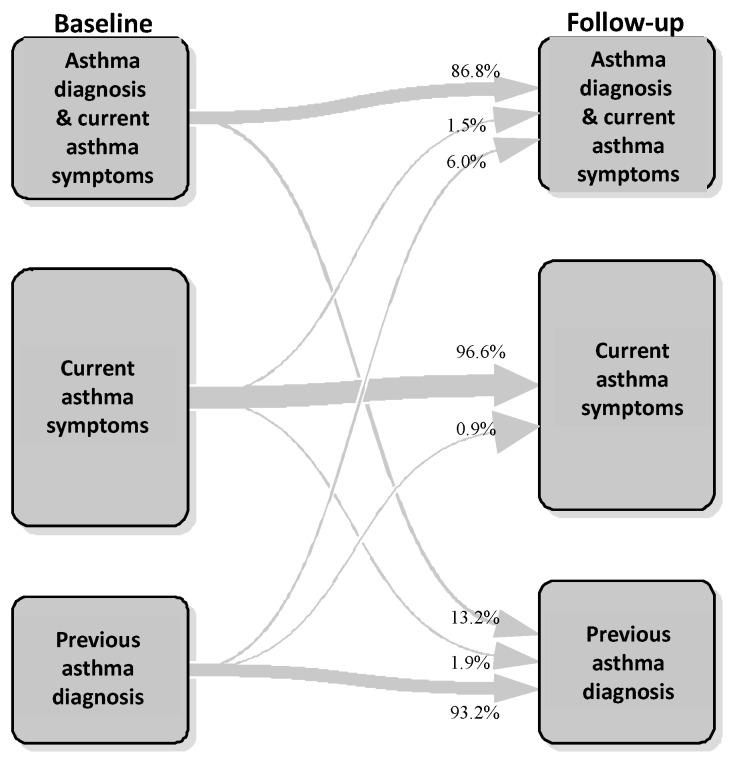
AGAVE phenotype transition plot.

**Figure 3 jcm-09-03632-f003:**
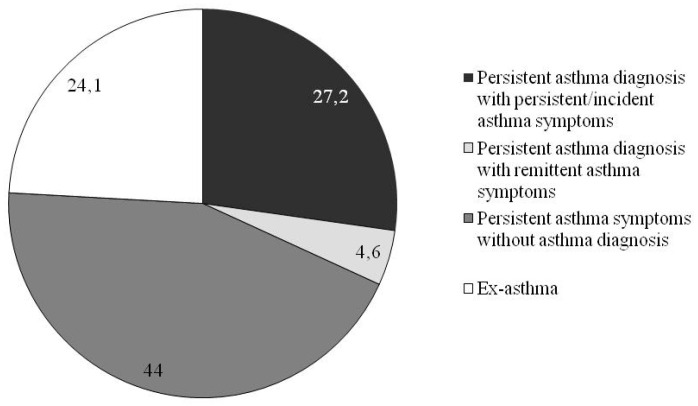
Longitudinal asthma patterns.

**Table 1 jcm-09-03632-t001:** Baseline descriptive characteristics of the investigated subjects (*n* = 452).

Gender (%):	
Males	47.3
Females	52.7
Age (mean ± SD ^1^) (years)	56.7 ± 15.5
Age range (min-max)	17–91
BMI ^2^ (kg/m^2^) groups ^3^ (%):	
obese (BMI ≥ 30 kg/m^2^)	15.7
overweight (BMI 25.0–29.9 kg/m^2^)	35.3
normal weight (BMI 18.5–24.9 kg/m^2^)	46.6
underweight (BMI < 18.5 kg/m^2^)	2.4
Educational level (%):	
elementary/junior high school	45.2
high school	35.9
university	18.9

^1^ SD, standard deviation; ^2^ BMI, body mass index; ^3^ threshold values recommended by WHO.

**Table 2 jcm-09-03632-t002:** Baseline host and environmental risk factors of the investigated subjects (*n* = 452).

***Host Risk Factors***	
Reported SPT ^1^ positivity (%)	74.9
Family history of asthma (%)	40.7
Allergic rhinitis (%)	40.8
GERD ^2^ (%)	29.6
Sleep apnea (%)	14.7
Recurrent respiratory infections (%)	13.5
COPD ^3^ (%)	12.5
Nasal polyps (%)	3.4
***Environmental Risk Factors***	
Traffic exposure at home address (%)	58.4
Smoking habits (%):	
current smokers	19.4
ex-smokers	40.3
Pack-years in current smokers (mean ± SD ^4^), n	23.6 ± 17.7
Pack-years in ex-smokers (mean ± SD ^4^), n	27.2 ± 27.9
Secondhand smoke exposure (%)	17.4

^1^ SPT, skin prick test (only subjects performing skin prick test *n* = 211); ^2^ GERD, Gastroesophageal reflux disease; ^3^ COPD, chronic obstructive pulmonary disease; ^4^ SD, standard deviation.

**Table 3 jcm-09-03632-t003:** Asthma symptoms/diagnosis frequency (%) within each cross-sectional phenotype.

	*Baseline*			*Follow-Up*		
	Asthma Diagnosis and Current Asthma Symptoms (28.5%)	Current Asthma Symptoms (45.6%)	Previous Asthma Diagnosis (25.9%)	Asthma Diagnosis and Current Asthma Symptoms (27.0%)	Current Asthma Symptoms (44.2%)	Previous Asthma Diagnosis (28.8%)
Current wheeze	64.8	30.2	12	73.8	32	9.2
Current asthma attacks	57.4	10.2	0	50.4	13.5	3.1
Current asthma diagnosis	85.3	0	0	86.1	1.5	0.8
Previous asthma diagnosis	14.7	0	100	11.5	0	99.2

**Table 4 jcm-09-03632-t004:** Baseline descriptive characteristics by longitudinal asthma patterns.

	Persistent Asthma Diagnosis with Persistent/Incident Asthma Symptoms (*n* = 123)	Persistent Asthma Diagnosis with Remittent Asthma Symptoms (*n* = 21)	Persistent Asthma Symptoms without Asthma Diagnosis (*n* = 199)	Ex-Asthma (*n* = 109)	*p*-Value
Sex (%):					0.108
males	39.0	52.4	52.8	45.9	
females	61.0	47.6	47.2	54.1	
Age (years) (mean ± SD ^1^)	**55.6 ± 16.8**	**47.9 ± 14.8**	**61.6 ± 14.1**	**50.6 ± 13.7**	**0.000**
BMI ^2^ groups (%):					**0.000**
obese	**15.4**	**4.86**	**19.1**	**12.0**	
overweight	**31.7**	**47.6**	**43.2**	**22.2**	
underweight/normal weight	**52.8**	**47.6**	**37.7**	**65.7**	
Educational level (%):					**0.000**
elementary/junior high school	**43.3**	**23.8**	**55.8**	**32.1**	
high school	**37.5**	**33.3**	**30.7**	**44.0**	
university	**19.2**	**42.9**	**13.6**	**23.9**	

^1^ SD, standard deviation; ^2^ BMI, body mass index. Statistically significant values are reported in bold.

**Table 5 jcm-09-03632-t005:** Baseline asthma-related indicators by longitudinal asthma patterns.

	Persistent Asthma Diagnosis with Persistent/Incident Asthma Symptoms (*n* = 123)	Persistent Asthma Diagnosis with Remittent Asthma Symptoms (*n* = 21)	Persistent Asthma Symptoms without Asthma Diagnosis (*n* = 199)	Ex-Asthma (*n* = 109)	*p*-Value
Age at asthma diagnosis (mean ± SD ^1^) ^2^	**27.1 ± 20.6**	**9.9 ± 11.6**	**---**	**13.8 ± 12.7**	**0.000**
Age at first asthma symptoms (mean ± SD ^1^) ^2^	**25.3 ± 20.3**	**8.0 ± 7.6**	**---**	**13.1 ± 12.6**	**0.000**
Family history of asthma (%)	**56.1**	**52.4**	**32.6**	**35.2**	**0.000**
Last 12-month asthma exacerbations ^3^ (%)	**15.6**	**5.9**	**4.9**	**0.0**	**0.002**
Last 12-month asthma hospitalizations ^3^ (%)	5.0	5.9	0.0	1.6	0.144
Last 12-month asthma ED ^4^ visits ^3^ (%)	4.2	5.9	1.2	1.6	0.473
Reported SPT ^5^ positivity (%)	*83.5*	*80.0*	*65.5*	*70.5*	*0.081*

^1^ SD, standard deviation; ^2^ performed only on subjects reporting lifetime asthma diagnosis; ^3^ performed only on subjects reporting lifetime asthma diagnosis or symptoms in the last 12 months; ^4^ ED, emergency department; ^5^ skin prick test (only subjects performing skin prick test *n* = 211). Statistically significant values are reported in bold; borderline values are reported in italic.

**Table 6 jcm-09-03632-t006:** Baseline host and environmental risk factors by longitudinal asthma patterns.

	Persistent Asthma Diagnosis with Persistent/Incident Asthma Symptoms (*n* = 123)	Persistent Asthma Diagnosis with Remittent Asthma Symptoms (*n* = 21)	Persistent Asthma Symptoms without Asthma Diagnosis (*n* = 199)	Ex-Asthma (*n* = 109)	*p*-Value
*Asthma comorbidities*					
Allergic rhinitis (%)	**61.0**	**66.7**	**28.3**	**35.8**	**0.000**
GERD ^1^ (%)	*30.1*	*23.8*	*34.8*	*20.6*	*0.067*
Sleep apnea (%)	**20.3**	**10.0**	**18.3**	**2.8**	**0.000**
Recurrent respiratory infections (%)	**23.1**	**4.8**	**11.3**	**8.3**	**0.003**
COPD ^2^ (%)	**21.1**	**9.5**	**11.9**	**4.6**	**0.002**
Nasal polyps	*5.7*	*0.0*	*4.1*	*0.0*	*0.074*
*Environmental risk factors*					
Traffic exposure at home address (%)	**69.1**	**38.1**	**56.3**	**54.1**	**0.015**
Smoking habits (%):					**0.000**
smokers	**15.4**	**14.3**	**27.1**	**11.0**	
ex-smokers	**34.1**	**28.6**	**47.2**	**36.7**	
nonsmokers	**50.4**	**57.1**	**25.6**	**52.3**	
Secondhand smoke exposure (%)	*18.9*	*9.5*	*21.4*	*10.2*	*0.066*

^1^ GERD, Gastroesophageal reflux disease; ^2^ COPD, chronic obstructive pulmonary disease. Statistically significant values are reported in bold; borderline values are reported in italic.

**Table 7 jcm-09-03632-t007:** Factors associated with longitudinal asthma patterns: Odds Ratio and 95% confidence intervals.

	Persistent Asthma Diagnosis with Persistent/Incident Asthma Symptoms	Persistent Asthma Diagnosis with Remittent Asthma Symptoms	Persistent Asthma Symptoms without Asthma Diagnosis
*Host factors*			
Allergic rhinitis	**3.12 (1.72–5.69)**	**3.38 (1.17–9.76)**	0.99 (0.56–1.78)
(ref: no)	1.00	1.00	1.00
COPD ^1^	**4.76 (1.60–4.16)**	3.48 (0.55–21.94)	1.32 (0.44–3.97)
(ref: no)	1.00	1.00	1.00
Sleep apnea	**5.99 (1.67–21.39)**	3.49 (0.50–24.44)	**5.32 (1.52–18.58)**
(ref: no)	1.00	1.00	1.00
*Environmental factors*			
Traffic exposure at			
home address	**1.86 (1.02–3.38)**	0.48 (0.17–1.35)	1.31 (0.75–2.28)
(ref: no)	1.00	1.00	1.00
Smoking habits:			
smokers	1.59 (0.64–3.96)	1.19 (0.26–5.47)	**6.24 (2.68–14.51)**
ex-smokers	0.71 (0.36–1.40)	0.45 (0.13–1.60)	1.71 (0.90–3.25)
(ref: non smokers)	1.00	1.00	1.00
Secondhand smoke exposure	**2.64 (1.10–6.33)**	0.93 (0.17–4.94)	**3.28 (1.41–7.66)**
(ref: no)	1.00	1.00	1.00
*Adjustment factors*			
Age (unit increase)	**1.02 (1.00–1.04)** **1.00**	0.98 (0.94–1.02) 1.00	**1.05 (1.03–1.07)** **1.00**
Males	0.84 (0.44–1.59)	1.49 (0.47–4.70)	0.84 (0.47–1.53)
(ref: females)	1.00	1.00	1.00
BMI categories:			
obese	1.46 (0.60–3.51)	0.63 (0.07–5.85)	1.79 (0.80–4.03)
overweight	1.77 (0.84–3.72)	**4.58 (1.33–15.72)**	**2.37 (1.20–4.68)**
(ref: underweight/normal weight)	1.00	1.00	1.00
Family history of asthma	**2.15 (1.18–3.92)**	*2.60 (0.91–7.45)*	0.91 (0.51–1.62)
(ref: no)	1.00	1.00	1.00

^1^ COPD, chronic obstructive pulmonary disease; reference category, ex-asthma; in italic, borderline values; in bold, statistically significant values. Adjusted for sex, age, BMI, and family history of asthma.

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
