# Peer review of "Longitudinal Asthma Patterns in Italian Adult General Population Samples: Host and Environmental Risk Factors"

_jcm, 2020, doi:10.3390/jcm9113632_

Round 1
Reviewer 1 Report
In this manuscript, the authors conducted phone interviews in 452 participants with asthma who were identified using previously conducted survey studies in nearby regions of Italy. The authors designed a custom questionnaire to probe asthma characteristics, and subsequently performed latent transition analysis to first identify cross-sectional phenotypes or groups in the total cohort and then determined the stability or any movement of participants within or between identified phenotypes. As the title suggests, the resultant clusters differed from each other based on host factors including age at asthma diagnosis, family history of asthma and exacerbation history, as well as environmental factors including comorbidities and smoke and traffic exposure.
The key strengths of this work are large longitudinal sample size of a real-world asthma sample and novel application of latent transition analysis. My main concerns are with the very minimal methodological information provided about the questionnaire used to interview participants, which makes it somewhat challenging to interpret the results.
Major Comments:
- More detail should be included in the methods for the data collection tool used here, please be more specific as to what was asked of participants. This can be kind of pieced together while reading the manuscript from start to finish but it should be very clear in the methods. What was the length of the phone interview with participants? What time frame were participants expected to recall for symptoms, etc.? Which questionnaires were used to derive the AGAVE questionnaire? What was the purpose for generating a new tool, that is not validated, rather than using already validated tools? Was the tool designed to specifically enable latent transition analysis? For reproducibility and generalizability purposes, consider including the entire questionnaire/tool in the online supplement to enable future research.
- I am not sure how much confidence there can be in the cross-sectional and longitudinal phenotypes without information about current asthma medications and asthma control. It is not clear whether these were actually probed using the custom design questionnaire, but they are not reported which makes me think it was not measured. How can the authors be sure ‘previous asthma diagnosis’ is not just well-controlled asthma using medications? Or, for those that transition from ‘previous asthma diagnosis’ to one of the other groups, that this is not related to an asthma exacerbation or loss of asthma control? Or, that ‘remission’ is not just a treatment change that improves asthma control?
- The target longitudinal time frame of the study is tucked away in the results, please include clearly in the methods. The target time was 12 months between baseline and follow-up phone interviews, but did the actual result differ once the study was completed? Was there a +/- time frame for the follow-up time points to be completed?
- I suggest showing other results from some of the variables in the questionnaire in Tables 4-6, even though they were not statistically significant. It appears somewhat suspicious with deliberating missing data, especially since it is not clear what exactly was measured using the questionnaire.
Minor Comments:
- Methods Study Population lines 70-75: Are all participants in these studies mutually exclusive or is there any overlap in the cohorts, for example longitudinal follow-up between PD1-2 and PI1-3? Figure 1 suggests they are mutually exclusive but please clarify in the text.
- Result Table 1: How was BMI determined with a phone interview? Were height and weight reported by the patient over the phone? If so, please address how patient reported height/weight may influence/bias the BMI results, for example patients reporting incorrect weight.
Reviewer 2 Report
General comments
The study used latent transition analysis (LTA) to detect baseline asthma phenotypes and 12-month follow-up longitudinal patterns in 452 adults (≥ 16 years) with an asthma diagnosis or asthma symptoms recruited to an online registry (AGAVE) using information obtained from telephone surveys. Three cross-sectional phenotypes and four longitudinal patterns were identified: uncontrolled asthma was associations with passive smoke and traffic exposure; persistent asthma symptoms were associated with active/passive smoke. The statistical approach (latent transition analysis) used to identify asthma phenotypes in a real-life population Italian is new and of interest. The limitations of the study are acknowledged by the authors, such as those associated with the use of a telephone survey and reporting bias. I’ve a few specific comments that are listed below.
Main specific comments
- Introduction
The authors should Include a brief statement explaining the reason for using latent transition analysis to identify asthma phenotypes. What the advantages over other analytical approaches? Reference previous studies in asthma that have used this statistical approach, for example, Boudier A et al. (2013). Ten-Year Follow-up of Cluster-based Asthma Phenotypes in Adults. A Pooled Analysis of Three Cohorts. Am J Respir Crit Care Med 188: 550-560; Weinmayr G, et al. (2013). Asthma phenotypes identified by latent class analysis in the ISAAC phase II Spain study. Clin Exp Allergy 43: 223-232.
- Page 9, lines 225-7: Rephrase sentence to state that the study population had asthma (ref 20)
- Page 9, line 231: omit ‘At last’
- Page 9, lines 252-253: Consider rephrasing the first sentence, such as ‘A significantly higher risk of persistent asthma symptoms without a lifetime diagnosis of asthma was found in current smokers (OR 6.24), which could indicate underdiagnosis of asthma in this group [33-34],
- Table 1: Provide definitions used to categorize BMI subgroups
- Table 2: Add ‘current’ before ‘smokers’ subgroup. If available, please provide information on pack year history
Round 2
Reviewer 1 Report
The revised manuscript is much improved and the clarity has been enhanced. Thank you for addressing my concerns and suggested revisions.